# Application of clown care in hospitalized children: A scoping review

Guo Xin[1]☯, Fu Yingping[1]☯, Chen Yue[1‡], Wu Jiaming[1‡], Hu Xue [2]*

1 School of Nursing, Yunnan University of Chinese Medicine, Kunming City, Yunnan Province, China,
2 Neonatology, Yan'an Hospital Affiliated to Kunming Medical University, Kunming City, Yunnan Province, China

☯ These authors contributed equally to this work.
‡ CY and WJ also contributed equally to this work.
* huxue123451@qq.com

## Abstract

### Purpose

This study aims to provide healthcare providers with insights into relevant interventions by examining the timing, format, content, outcome measures, and effects of clown care interventions on hospitalized children.

### Methods

This study utilized a scoping review methodology based on the framework by Arksey and O'Malley. The search for Chinese and English literature on the utilization of clown care for hospitalized children was conducted in databases including CNKI, VIP, Wanfang, China Biology Medicine Database, PubMed, Embase, Web of Science, CINAHL, and Cochrane Library. At the same time, the references and other grey literatures included in the paper were also manually searched. covering publications from inception to December 10, 2023. Screening of the literature was done based on predefined inclusion and exclusion criteria, with data extraction and synthesis carried out independently by two researchers.

### Results

Out of 1084 articles screened, 18 were included. Clown care can be used with children who have burns, perioperative care, fracture rehabilitation, autism, puncture therapy, chemotherapy, respiratory pathologies and disabled children, according to a review of the literature. Clown care is feasible and available for these kids, and it has been shown to have favorable impacts. The main interventions used in clown care are guided imagery, cognitive coping, emotional reflection, distraction, and empowerment. Physical(reduced children's pain levels), psychological(reduced anxiety, improved coping skills, quicker recovery), cognitive-behavioral(for example, the number of meaningful words that children with autism say in 10 minutes.), and other indications are the several types of clown care outcome indicators.

**Data Availability Statement:** All relevant data are within the manuscript and its Supporting information files.

**Funding:** The author(s) received no specific funding for this work.

**Competing interests:** The authors have declared that no competing interests exist.

## Conclusions

Clown care can lessen a hospitalized child's pain, alleviate their worry, and divert their focus from illness and suffering. However, not all pediatric patients benefit from clown care. Every youngster is an independent individual with unique personality traits, and some may experience intense pain along with a terror of the clown role. Future research should concentrate on understanding the characteristics of hospitalized children to develop tailored clown care programs that can enhance clinical effectiveness.

## Trial registration number

https://doi.org/10.17605/OSF.IO/B4KMV.

## Introduction

Hospitalization can have adverse effects on an individual's emotional, behavioral, cognitive, and educational development, particularly in children. [1–4] due to their limited cognitive abilities and unfamiliarity with the hospital environment. The youngster may not be able to handle the unpleasant hospital stay due to the underdeveloped brain and nervous system, the absence of a defined objective, and the agony brought on by the illness. The child's surrounding environment is exceedingly sensitive, leading to feelings of fear when confronted with the sight of medical staff in white gowns. Additionally, prolonged exposure to an unfamiliar setting, along with continuous diagnostic and nursing interventions, gives rise to anxiety, depression, and other detrimental emotions. These feelings are expressed through behaviors such as crying and resistance, which negatively impact the child's willingness to comply with medical directives and hinder the effective progression of clinical diagnosis and treatment.

At present, a variety of non-drug intervention methods are used to relieve children's pain and anxiety associated with medical operations [5, 6], and among these methods, clown care has demonstrated a significant effect. Clown care, also known as clown therapy or medical clowning, has emerged as a popular non-pharmacological intervention in pediatric care, in recent years. Clown care is a non-drug intervention method based on therapeutic humor. It is mainly manifested in the following four aspects, from a physiological perspective, patients can release endorphins through humor and hearty laughter, thereby regulating and stimulating their immune response. Cognitively, clowns embody various roles that divert patients' attention from their conditions, fostering imagination and facilitating self-expression. In terms of emotional well-being, patients are encouraged to substitute negative feelings with positive ones. Socially, clowns engage with patients to enhance interaction and connection. It is upon this foundation of humor theory that clown care is recognized for its valuable contributions within medical environments [7]. In the early 20th century, two clowns first appeared in a paediatric ward in London [8]. In the 1980s, Michael Christensen, a professional clown, founded the Clown Care Center in New York [9]. In the same year, Canadian child life expert Karen Ridd established the Clown Care Program at Winnipeg Hospital [10]. Since then, clown care has been formally integrated into the medical and nursing process. Hospital clowning continues to grow around the world. Numerous hospitals across the United States, Canada, Europe, Israel, Australia, South Africa, Asia, and South America presently implement clown intervention programs [11].

Research has demonstrated that therapeutic clowning can significantly alleviate anxiety and fear in hospitalized children, mitigate postoperative pain, and positively contribute to enhancing the health outcomes during children's disease rehabilitation [4]. Nevertheless, clown therapy may not be appropriate for every hospitalized child. The age of participants was found to be a significant determinant in treatment adherence studies, according to a systematic review [12]. Adolescents may exhibit greater behavioral control and a tendency to conceal their emotions, whereas younger children are more likely to display emotional distress readily. Compared to younger children, teens may benefit less from clown care. An additional study [13] revealed that children's distress was unaffected by their interactions with clowns. This inconsistency could be due to methodological reasons. Clown care programs are still in the early stages of development in China, primarily due to cultural differences [14], the Western archetype of clowns, characterized by red-nosed masks and outlandishly dramatic costumes costumes, may not be appropriate for Chinese medical facilities. Incorporating a traditional Chinese cartoon character, like the Monkey King, could resonate more positively with children in hospital settings. In contrast to other countries, the implementation of therapeutic clowning for hospitalized children in China has been limited in duration. The majority of "clown doctors" are volunteers, which results in a lack of an organized training framework. Additionally, there is no standardized procedure regarding the recruitment criteria, sources, conditions, training methodologies, content, or length of training for "clown doctors."

Many reviews have clarified the effectiveness of the application of clown care in pediatric patients, but there is a lack of comprehensive descriptions of the intervention timing, form and content of intervention, and outcome indicators of existing clown care. A scope review can summarize a range of documentary evidence to show the breadth and depth of a knowledge domain. Therefore, guided by the scope review report framework [15], this study summarized and analyzed the relevant literature in this field at home and abroad, and summarized the intervention timing, form and content of intervention, outcome indicators, and effects of clown care in hospitalized children, with a view to providing references for medical staff to carry out relevant research. This report follows the PRISMA-ScR checklist (S1 File).

## Methods

This study follows the framework outlined by Arksey and O'Malley, which includes five essential stages: 1) Formulating the research question; 2) Identifying pertinent studies; 3) Selecting studies; 4) Organizing data, and; 5) Synthesizing, summarizing, and presenting findings.

### Stage one: Formulating the research question

According to the principle of participants, concept, context, the study object is hospitalized children, the study concept is clown care, and the study context is the application of clown care in hospitalized children.

Finalizing research questions:

- What are the advantages and limitations of clown care in the application of hospitalized children?

- What are the factors influencing the application of clown care in hospitalized children?

- How can existing programs and application modalities aid hospitalized children in adapting to their new environment and alleviating negative emotions?

**Table 1. Pilot search in PubMed electronic database.**

| Date | Keywords | Search results |
|---|---|---|
| 10/12/2023 | #1Clown [Title/Abstract] OR medical clown [Title/Abstract] OR clown doctor [Title/Abstract] OR clown intervention [Title/Abstract] OR clown therapy [Title/Abstract] OR clown nurse [Title/Abstract]<br>#2child [Mesh Terms] OR children [Title/Abstract] OR pediatric [Title/Abstract] OR kids [Title/Abstract]<br>#3 #1AND#2 | 1084 |

## Stage two: Identifying relevant studies

A systematic search was conducted across databases such as CNKI, VIP, Wanfang, China Biology Medicine Database, PubMed, Embase, Web of Science, CINAHL, and Cochrane Library, aligning with the identified research queries. The search was restricted to articles published up to December 10, 2023. A comprehensive search strategy combining controlled vocabulary and free-text terms with Boolean operators "AND" and "OR" was employed. The search strategy tested in PubMed is presented in Table 1 (S2 File). Prior to screening, systematic training exercises were conducted on researchers to ensure the consistency of the studies. When there are discrepancies between two individuals regarding the literature selection process, the viewpoint of a third researcher is sought to ensure that no essential literature is overlooked in the inclusion procedure. All related articles are managed using EndNote X9 Reference Manager, which also helps to identify duplicate entries.

## Stage three: Study selection

**Eligibility and inclusion criteria.** *Inclusion criteria.*

**(1)Participant (P)**
   Hospitalized children<18 years of age, along with any accompanying family members;

**(2)Interest of phenomena (I)**
   Receiving hospital clowns intervention;

**(3) Comparison (C)**
   Compared receipt of hospital clowns to standard of care;

**(4)Outcome (O)**
   The application and effect evaluation of clown care in hospitalized children;

**(5)Study design (S)**
   Original research literature, encompassing qualitative studies, case-control studies, cohort studies, case studies, etc.

   *Exclusion criteria.*

1. Literature where the effect of clown care in combination with other interventions is challenging to quantify;

2. Conference papers or incomplete texts with ambiguous or imprecise data;

3. Duplicate publications;

4. Literature not available in Chinese or English.

**Selection process.** To eliminate duplicates, all literature was imported into the EndNote X9 software. Two researchers trained in evidence-based practice independently screened titles and abstracts based on the inclusion and exclusion criteria. Full-text review was conducted for all potentially relevant articles. In case of disagreements, a third researcher reviewed the material and a consensus was reached. The screening process is depicted in S1 Fig.

## Stage four: Charting the data

A form be developed in Excel for the data extraction and piloted to ensure its accuracy. After carefully reading the entire text, the first and second researchers collected all pertinent data from the included literature. The following information be included in the data extraction form: the study's title, authors, nation, publishing date, kind of study, sample size, study population, intervention strategy, and outcome indicators. Once the data were plotted, the third researcher, through repeated reading and analysis of the included literature, checked the extracted data and results, and updated the extracted data as needed, so as to obtain all the data to make the study more complete and form the final results.

## Stage five: Collating, summarizing and reporting results

The following be serve as the foundation for themes pertaining to the study objectives: (i)What types of illnesses does clown care predominantly address in pediatric inpatients?(ii) How is clown care applied in pediatric settings, and which patients benefit most? (iii) How does providing clown care to pediatric patients in hospitals affect their evaluation? This will function as a story with pertinent themes.

## Results

### Characteristics of the studies selected

An initial search yielded 1,082 articles. After deleting duplicate articles, 429 articles were obtained. Further reading of the titles and abstracts of these articles yielded 87 articles. Finally, after reading the full text, 16 articles were obtained after excluding those that did not meet the standard. 2 articles were obtained by other means. These articles were selected by excluding those that did not fit the content and topic of the study substances, as well as those that cannot be accessed in full text. Finally, a total of 18 articles were included. At least two reviewers screened each result by title and abstract, with disagreements resolved by discussion with the review team. The remaining full text papers were accessed and screened against the eligibility criteria, again by at least two reviewers with disagreements resolved by discussion. Among these, 13 were in English and 5 were in Chinese. The included studies originated from countries such as Italy [16, 17, 23, 25, 31], Israel [18, 30], Denmark [19], Brazil [29], Germany [24], the United States [20], Turkey [12], and China [21, 22, 26, 27, 28], Canada [32] are among the countries whose works were published between 2005 and 2023. Table 2 (S1 Table) displays the fundamental features of the included literatur.

### Essential elements of clown care

**Clown care clients and clown care practitioners.** Clown care is utilized in various settings for hospitalized children, including puncture therapy [16–22], perioperative [23–26], burn treatment [12, 27], fracture rehabilitation [28], chemotherapy support [29], autism intervention [30], respiratory pathologies [31], and disabled children [32]. Evaluation of clown care often involves measuring parent satisfaction, caregiver anxiety levels, and caregiver trait anxiety in the context of hospitalized children [18, 21–23, 25, 27, 28, 31]. The evaluation of

**Table 2. General characteristics of the included studies (n = 18).**

| Reference | publication year | Country | Study Design | population | clowns practitioner | Timing of intervention | Forms of intervention | conclusion mark |
|---|---|---|---|---|---|---|---|---|
| Ben-Pazi H, et al. [16] | 2017 | Italy | Randomized controlled trial | A | Medical clowns | puncture therapy | Cognitive responses; Empowerment; Emotional reflection; | ① |
| Felluga M, et al. [17] | 2016 | Italy | quasi-randomized controlled trial | A | Medical clowns | puncture therapy | Cognitive responses; Divert attention; | ①② |
| Goldberg A, et al [18] | 2014 | Israel | Randomized controlled tria | A | Medical clowns | puncture therapy | Divert attention | ①② |
| Kristensen HN, et al. [19] | 2018 | Denmark | Controlled clinical trial | A | Medical clowns | puncture therapy | Cognitive responses; Divert attention; | ① |
| Meiri N, et al.[20] | 2015 | United States | Randomized controlled tria | A | Medical clowns | puncture therapy | Divert attention; | ①②④ |
| NingXuemei,et al. [21] | 2023 | China | Randomized controlled tria | A | Medical clowns | puncture therapy | Divert attention | ①②④ |
| Yang Fan, et al. [22] | 2022 | China | Randomized controlled tria | A | Medical clowns | puncture therapy | Cognitive responses; Divert attention; Guided imagery; | ①④ |
| Dionigi A, et al. [23] | 2014 | Italy | Randomized controlled tria | B | Medical clowns | perioperative period | Divert attention; | ① |
| Markova G,et al. [24] | 2021 | Germany | Randomized controlled tria | B | Medical clowns | perioperative period | Guided imagery; Divert attention; | ② |
| Vagnoli L, et al. [25] | 2005 | Italy | Randomized controlled tria | B | Medical clowns | perioperative period | Guided imagery; Divert attention | ② |
| Cheng Zongyan, et al. [26] | 2019 | China | Randomized controlled tria | B | Medical clowns | perioperative period | Cognitive responses; Divert attention; Guided imagery; | ①④ |
| Yildirim M, et al. [12] | 2018 | Turkish | Randomized controlled tria | C | Medical clowns | dressing changes | Guided imagery; Emotional reflection; | ③ |
| Yan Guifan, et al. [27] | 2023 | China | Randomized controlled tria | C | Medical clowns | dressing changes | Divert attention; | ①④ |
| Wan Lin, et al. [28] | 2023 | China | Randomized controlled tria | D | Medical clowns | Postoperative fracture and rehabilitation | Divert attention; Cognitive responses | ①④ |
| Lopes-Junior LC, et al [29] | 2020 | Brazil | quasi-randomized controlled trial | E | volunteers | Before and after chemotherapy | Divert attention; | ②④ |
| Shefer S, et al. [30] | 2019 | Israel | Controlled clinical trial | F | Medical clowns | When treating autism | Emotional reflection; Cognitive responses | ③ |
| Bertini, et al. [31] | 2011 | Italy | Randomized controlled tria | G | Medical clowns | When the child was in the hospital room | Divert attention | ④ |
| Shauna, et al.[32] | 2011 | Canada | non-randomised controlled trials | H | Medical clowns | When the child was in the hospital room | Divert attention Empowerment; | ② |

A children with puncture therapy, B children in perioperative period, C children with burns, D children after fracture surgery, E children with chemotherapy, F children with autism;G Children hospitalized for respiratory pathologies, H disabled children. ①physiological indicators, ②psychological indicators, ③ cognitive-behavioral indicators, ④other indicators.

caregiver satisfaction primarily involved the use of questionnaire surveys or follow-up assessments of healthcare personnel. This methodology allows for a swift and straightforward capture of the subjective experiences reported by the parents of hospitalized children, making it both practical and accessible. However, it is important to note that this approach may lack the precision and objectivity associated with standardized assessment scales.

Individuals responsible for designing and implementing clown care programs are referred to as clown care practitioners in this study. These practitioners primarily consist of medical clowns who have received training in sociology, psychology, pedagogy, acting, nursing, and healthcare [12, 16–28, 30, 31, 32]. Additionally, clown care volunteers [29], including clown doctors, clown nurses, and professional medical clowns, are also included within this category. However, there are no uniform requirements for "clown doctors." Only the basic rules of public morality were laid down. For example, the physical health of each "clown doctor" must be ensured, and the clown doctor must comply with the relevant regulations and codes of conduct of the medical institution. Patients must be served impartially, regardless of their gender, age, beliefs, or illness. There is no uniform standard for the recruitment methods, sources, conditions, training methods, content, and duration of "clown doctors," which lacks standardization.

**Timing of clown care intervention for hospitalized children.** Among the sixteen studies compiled, seven studies [16–22] discussed the application of clown care in puncture therapy, encompassing procedures like venepuncture, intravenous infusions, phlebotomy, botulinum toxin injections, and allergy prick testing conducted during the hospital stay. Four publications [23–26] focused on the use of clown care in anesthesia for induction, postoperative care, and preoperative care in the perioperative period. Two studies [12, 27] examined discomfort related to burns on the body, emphasizing pre-procedure anxiety, discomfort during dressing changes, and post-procedural relaxation. Postoperative rehabilitation following a fracture was explored in one paper [28], covering both postoperative fracture management and rehabilitation. Another paper [29] discussed the application of clown care during chemotherapy sessions. Additionally, one study [30] investigated the provision of clown care to children with autism, focusing on the timing of interventions ten minutes before, during, and after the procedure. The last two articles introduced the application effect of clown care in hospitalized children with respiratory diseases [31] and disabled children [32], respectively.

**Form and content of clown care.** The elements of clown care can be encapsulated by terms such as "cognitive coping," "guided imagery," "distraction," "emotional reflection," and "empowerment." "Cognitive coping" [16, 17, 19, 22, 26, 28, 30] is designed to encourage pediatric patients to actively respond to the challenges brought by the disease and improve their self-cognitive ability. Studies [33] have shown that "medical clowns" improve their self-cognitive ability by guiding perioperative pediatric patients to "Give yourself a like" other positive behaviors to improve their self-cognition ability. "Guided imagery" [12, 16, 22, 24–26, 29] entails using clowning to create positive memories and help children cope with pain by guiding them to imagine pleasant experiences. Divert attention" [12, 17–29, 31, 32] pertains to the use of medical clowns who engage with children through enjoyable experiences designed to capture their attention. These interactions can be tailored to the child's age and may include magic tricks, pantomime, storytelling, and music, among other forms of entertainment. For instance, adolescents might appreciate magic performances or juggling acts, while younger children may enjoy fairy tales or their favorite children's songs. By providing customized entertainment programs aligned with the age and interests of hospitalized children, we can more effectively alleviate discomfort and anxiety during their stay, thereby diverting their attention from distressing circumstances. "Emotional reflection" [12, 16, 26, 30] involves adapting the clown's performance to the child's emotional cues, engaging with the child in a

manner that resonates with their feelings, and encouraging the child to express themselves during the interaction. "Empowerment" [16, 32] involves the clown adjusting their actions based on the child's input, fostering the child's confidence and agency. By simulating medical procedures as a game, children gain a better understanding of the process, reducing anxiety and fear.

## Outcome indicators and effects

The collection of 18 studies outlined five sets of outcome measures: behavioral, cognitive-behavioral, physical, and miscellaneous markers. Physiological indicators were reported in eleven studies [16–22, 26, 27, 28, 31] focusing primarily on children's pain levels (measured using various scales such as Pain Visual Analog Scale, Pain Ruler, Numeric Pain Rating Scale, Facial Expression Pain Inventory, FLACC Pain Assessment Scale, self-reported pain levels, modified Facial Pain Expression Scale known as Piggyback), children's fatigue (assessed using Pediatric Quality of Life Scale, Child Multidimensional Fatigue Scale), duration of children's crying, healing times for fractures (using Femadez-esteve scale), and improvement in joint function (measured with Flynn scale).Psychological markers were reported in eight studies [17, 18, 21, 23–25, 29, 32] focusing on children's positive emotions, caregiver anxiety levels (State-Trait Anxiety Inventory), children's emotional states and anxiety levels (Modified Yale Preoperative Anxiety Scale), trait anxiety (Children's State-Trait Anxiety Inventory), and psychological stress (Children's Stress Inventory). This study's seven included studies demonstrated that, when compared to conventional care, children and adolescents who had hospital clowns—whether or not a parent was present at the time of intervention—reported much less anxiety and improved psychological adjustment. Cognitive-behavioral indicators were discussed in three studies [12, 27, 30], highlighting metrics such as the number of meaningful words spoken in ten minutes, instances of social smiles (Autism Diagnostic Scale, Second Edition), interactions during a game of "catch," and compliance with medication regimens. Additional measures, including salivary cortisol levels, duration of child's crying, amount of pain medication used within 24 hours post-surgery, success rate of medical procedures, and parental satisfaction (assessed through questionnaires), were detailed in nine studies [12, 20–22, 26–29, 31].

## Discussion

### Clown care interventions are varied, but programs lack individualization

The study findings suggest that while there exists a wide range of clown care interventions, the predominant approaches revolve around "cognitive coping," "guided imagery," "distraction," "emotional reflection," and "empowerment." Each patient is a unique individual with distinct characteristics. Clown phobia refers to the extreme aversion and distress experienced by individuals towards clown figures. A study of 95 patients with clown phobia found that clown phobia occurs in younger female children clownphobia were in female youngsters. Serious psychological discomfort, such as anxiety or panic attacks, as well as a decline in social function, can also contribute to the development of clownphobia. In a study involving 1160 children receiving clown care interventions, 14 of them exhibited fear or phobia towards medical clowns [33]. Fear of clowns can affect children at any age (range 1–15), any ethnicity, religion, or degree of illness. Further large scale studies are required to better understand this unique phenomenon of fear of clowns.

Theories of the aetiology of clown fear can be broadly split into three general categories: those relating to their physical appearance, those relating to their behaviour and those derived from learning and/or experience. Firstly, considering fear of clowns as deriving from aspects

of physical appearance, Moore [34] suggests that coulrophobia may stem from the uncanny valley effect which describes the feelings of eeriness and repulsion triggered by near human-looking objects. With regard to clowns, the distortion of their facial features through makeup gives them a 'near-human' quality which may elicit the uncanny valley effect. Secondly, behavioural explanations of clown fear focus on the unpredictability of their behaviour. Clowns laughing at random intervals, as well as the nature of their performances emphasizing magic and trickery, may make the child feel uncomfortable or upset. Finally, the fear of clowning may stem from the negative portrayal of clowns in the media, as well as from negative experiences in the same year. Research indicates that symptoms of clown phobia can persist for up to 60 years [35]. Furthermore, a report on clinical characteristics from clown care indicated that 9.5% of patients experienced no notable enhancement in their fear of clowns, nor did they find alleviation from social or psychological issues. Children may lack the cognitive capacity, and their hospitalization can already induce significant stress. Incorrectly tailored clown care interventions based on the children's individual traits could lead to lasting psychological harm and hinder the normal development of their psychological, physiological, and cognitive functions. To enhance the efficacy of clown care programs and tailor them more effectively, it is crucial to integrate the unique needs and characteristics of hospitalized children into these interventions. Clinical practitioners should assess high-risk children for clown phobia before implementing clown care to prevent potential harm and address underlying issues. Further research is needed to enhance the focus, effectiveness, and clinical relevance of clown care for hospitalized children. Currently, there is a lack of assessment tools specifically designed for pediatric clown care.

## A lesser known study in clown care examines the effect on biochemical tests such as salivary cortisol concentration in hospitalized children

To effectively manage and alleviate the pain experienced by hospitalized children, it is essential to conduct a thorough and precise evaluation of their discomfort. As some hospitalized children may struggle to articulate their pain levels, healthcare professionals often rely on observational scales to assess pain based on the children's behaviors, such as limb movements and facial expressions [36]. However, it has been suggested [37] that pain assessment tools focusing on behavioral and facial cues might be limited to subcortical somatic and autonomic motor pathways, which may not accurately capture a child's pain experience. Cortisol, the primary stress hormone produced by the body, has emerged as a valuable objective marker for evaluating children's discomfort and the effectiveness of treatments [38]. When the body encounters pain stimuli, it elicits a corresponding neuroendocrine stress response, leading to an elevation in cortisol levels. Morelius et al. [39] demonstrated that painful interventions, such as fundus examinations and heel blood draws, can elevate cortisol levels in neonates, while certain non-pharmacological interventions may mitigate these cortisol levels. Cortisol secretion exhibits a circadian rhythm. Human cortisol metabolism adheres to a 24-hour physiological cycle, peaking between 6:00 and 8:00 AM, followed by a gradual decline, with the lowest levels occurring from midnight to 2:00 AM the subsequent day. Ivars et al. [40] undertook a 12-month study measuring salivary cortisol in healthy full-term and preterm infants, revealing that the salivary cortisol rhythm is established during the neonatal phase (one month postpartum), characterized by higher levels in the morning compared to the evening, which then stabilize over time. To mitigate the effects of circadian rhythm, clinical sampling should be conducted at the time of maximum cortisol concentration. Cortisol levels can be measured through blood, saliva, and urine samples. Saliva samples are easy to collect and non-invasive, and salivary cortisol represents biologically active free cortisol [41], enhancing the precision of assessments.

Among the 18 studies included in this investigation, only one [29] utilized salivary cortisol as an outcome measure to assess pain in children.

Currently, the utilization of salivary cortisol for assessing neonatal pain predominantly centers on planned pain responses. There is a scarcity of research investigating variations in salivary cortisol levels in newborns following non-pharmacological therapies. This gap is largely attributed to the intricate nature of endocrine dynamics, which may have interplay among different hormones. Research conducted by Pavlyshyn et al. [42]. indicates that interventions such as skin-to-skin contact can elevate oxytocin levels while concurrently diminishing salivary cortisol levels, thus enhancing stress responses in preterm infants. Both endorphins and oxytocin possess analgesic and pleasurable effects and serve as anti-stress hormones, whereas cortisol is regarded as the primary hormone associated with stress. Further research is necessary to delve deeper into the interactions between hormones, pain, and their underlying mechanisms. To provide scientific validation for the advancement of clown care programs that effectively alleviate pain, reduce physical and psychological distress, shorten hospital stays, and aid in recovery, integrating pain assessment scales with biochemical testing in future research and clinical applications can offer valuable insights.

## Exploring cost-effective and convenient alternatives to on-site clown care provision

The implementation of clown care clinical activities for hospitalized children was carried out on-site in the 18 publications included in this study. The healthcare landscape is evolving, with increasing demands on healthcare professionals as societal advancements and healthcare models undergo updating and transformation [30]. Pediatric wards entail high-risk, high-intensity, and high-pressure medical work compared to other departments. Introducing clown care may add to the workload of medical professionals. Moreover, the intense stress of clinical work could potentially lead to professional burnout and impact clinical outcomes for children [43]. Conversely, establishing a clown care team requires a substantial number of healthcare experts; therefore, virtual reality can be leveraged to enhance clinical tasks related to clown care interventions [44]. This approach can aid in the efficient allocation of medical resources, potentially leading to a more equitable distribution of human resources among clinical healthcare specialists. In addition to alleviating the burden on medical staff and enhancing treatment adherence for hospitalized children, virtual reality (VR) technology presents itself as a cutting-edge, cost-effective, and practical means to deliver care to children in hospital settings. By integrating virtual reality technology with clown care, such as offering a virtual reality interactive platform featuring children's favorite cartoons to aid in engagement, we can provide a more immersive and tailored care experience. The utilization of virtual reality technology in healthcare is expanding and has shown promise in various domains, including pain management [45], psychotherapy [46], and other areas. It is recommended that future research explores more affordable and convenient programs that leverage virtual reality technology to enhance care delivery.

## Clown care should be culturally appropriate and localized

The traditional Western clown appearance, characterized by white faces, red noses, and black eyes, may not be suitable for hospitalized children in China and many other Eastern countries due to cultural differences, potentially causing fear instead of comfort. Chinese children in hospitals may not respond positively to clown care interventions that involve mime, puppetry, or similar performance-based techniques. While some studies have demonstrated the benefits of clown care in alleviating pain and anxiety in children from diverse cultural backgrounds [46], it is evident that children from varying cultural backgrounds may respond differently to

the same clown care approaches, underscoring the influence of cultural values and backgrounds on the effectiveness of such interventions. Therefore, it is essential to carefully consider and incorporate the cultural values and norms of each nation when implementing clown care for hospitalized children from different cultural backgrounds. Developing a culturally localized clown care strategy specific to our country is imperative.

## Limitations of the study

Acknowledging the constraints of a scoping review is crucial. To begin with, the process of selecting articles did not involve any quality criteria. While this is not a mandatory methodological requirement for conducting a scoping review, it does hinder the authors' ability to evaluate the reliability and strength of the literature included, consequently impacting the confidence in the conclusions drawn from it. Additionally, the generalizability of the findings is restricted as we solely incorporated published works in both Chinese and English during the selection of literature for inclusion.

## Implications

This study delved into the scope of clown care initiatives for pediatric patients in hospitals, systematically categorizing research subjects, practitioners, intervention modalities, content, and outcome measures associated with clown care. Despite the existence of diverse clown care interventions, they lacked personalization. Additionally, the utilization of physiological markers such as salivary cortisol was infrequent, with measures and questionnaires being the predominant assessment tools for evaluating the impact of clown care. In comparison to the observational evaluation of behaviors like facial expressions and body language, cortisol serves as the primary hormone indicative of stress in the body. Its collection method is straightforward and non-invasive, which can make the detection results more accurate. Given the cultural disparities between China and Western nations, it is essential for Chinese medical institutions to develop an optimal clown care implementation strategy for pediatric patients in China. This should be informed by insights gained from international clown care practices while being tailored to align with the unique social and cultural context of China. To establish a robust and scientifically grounded framework for the clinical advancement of high-quality clown care programs for hospitalized children, future research should explore the influence of various cultural backgrounds and values.

## Limitations of the study

Acknowledging the constraints of a scoping review is crucial. To begin with, the process of selecting articles did not involve any quality criteria. While this is not a mandatory methodological requirement for conducting a scoping review, it does hinder the authors' ability to evaluate the reliability and strength of the literature included, consequently impacting the confidence in the conclusions drawn from it. Additionally, the generalizability of the findings is restricted as we solely incorporated published works in both Chinese and English during the selection of literature for inclusion.

## Supporting information

**S1 File. PRISMA-ScR checklist.**
(DOCX)

**S2 File. Search strategy.**
(DOCX)

**S1 Fig. Flow chart of literature screening.**
(JPG)

**S1 Table. General characteristics of the included studies.**
(DOCX)

## Author Contributions

**Conceptualization:** Fu Yingping.

**Data curation:** Guo Xin, Chen Yue, Wu Jiaming.

**Formal analysis:** Guo Xin, Fu Yingping, Chen Yue, Wu Jiaming, Hu Xue.

**Methodology:** Guo Xin, Fu Yingping.

**Project administration:** Guo Xin, Fu Yingping, Hu Xue.

**Supervision:** Guo Xin, Fu Yingping, Hu Xue.

**Writing – original draft:** Guo Xin, Fu Yingping, Chen Yue, Wu Jiaming, Hu Xue.

**Writing – review & editing:** Guo Xin, Fu Yingping, Chen Yue, Wu Jiaming, Hu Xue.

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
