## [Decision Letter · Decision Letter 0]

10 Oct 2024

PONE-D-24-35005Application of clown care in hospitalized children: A scoping reviewPLOS ONE

Dear Dr. Hu,

Thank you for submitting your manuscript to PLOS ONE. After careful consideration, we feel that it has merit but does not fully meet PLOS ONE’s publication criteria as it currently stands. Therefore, we invite you to submit a revised version of the manuscript that addresses the points raised during the review process.

We look forward to receiving your revised manuscript.

Kind regards,

Cho Lee Wong, PhD

Academic Editor

PLOS ONE

Journal Requirements:

2 Please include captions for your Supporting Information files at the end of your manuscript, and update any in-text citations to match accordingly. Please see our Supporting Information guidelines for more information: http://journals.plos.org/plosone/s/supporting-information.

3. We note that your Data Availability Statement is currently as follows:” All relevant data are within the manuscript and its Supporting Information files.”

Additional Editor Comments:

Thank you for the opportunity to read this manuscript. This topic touches on an important subject and is interesting.

However, a few comments for authors consideration:

Introduction:

- There are some reviews about medical clowning for sick children, how does this article differ from them?

- Why conduct a scoping review rather than other types of reviews?

Methods:

- For Eligibility criteria: Please state in PICOS.

- A similar meta-analysis review was conducted and completed in November 2019 and they yielded 14 studies. However, your review is a scoping review and should include more papers. Please revisit if you missed any potential research.

Discussion:

The discussion addresses the key points and limitations of the current study. However, it could be strengthened by 1) a more explicit comparison of the findings with the existing literature. 2) elaborate the implications for practice and policy; 3) provide more specific recommendations for future research.

Reviewers' comments:

Reviewer's Responses to Questions

**Comments to the Author**

1. Is the manuscript technically sound, and do the data support the conclusions?

Reviewer #1: Yes

Reviewer #2: Yes

2. Has the statistical analysis been performed appropriately and rigorously? 

Reviewer #1: Yes

Reviewer #2: Yes

3. Have the authors made all data underlying the findings in their manuscript fully available?

Reviewer #1: Yes

Reviewer #2: Yes

4. Is the manuscript presented in an intelligible fashion and written in standard English?

Reviewer #1: Yes

Reviewer #2: Yes

5. Review Comments to the Author

Reviewer #1: - On abstract please mention the time period

- C clearly state your research difference from other similar studies in 2023 and 2024(https://doi.org/10.1007%2Fs12519-023-00720-y, and https://doi.org/10.3389%2Ffped.2024.1324283)

Reviewer #2: Dear Authors,

Thank you for submitting your manuscript titled "Application of Clown Care in Hospitalized Children: A Scoping Review." The study presents valuable insights into the role of clown care interventions and is commendably structured. However, I believe there are several areas that could benefit from further enhancement. I encourage you to review the attached documents, which outline specific comments and suggestions. Addressing these points will significantly strengthen your manuscript.

Best regards,

6. PLOS authors have the option to publish the peer review history of their article (what does this mean?). If published, this will include your full peer review and any attached files.

Reviewer #1: **Yes: **Yideg Abinew

Reviewer #2: **Yes: **Seid Muhumed Abdilaahi, Departments of Pediatrics and Child Health Nursing, Jigjiga University, Jigjiga, Ethiopia

---

## [Author Response · Author response to Decision Letter 0]

25 Oct 2024

Dear Editor and Reviewers:

Greetings!

First of all, on behalf of all the authors, please allow me to extend our sincerest thanks to you and the reviewers. We would like to thank you for taking the time out of your busy schedules to review our paper in detail and provide valuable comments and suggestions. We are fully aware of the importance of these comments in improving the quality of the paper and ensuring academic rigor.

After receiving the review comments, we immediately organized our team to conduct in-depth discussion and analysis, and revised and improved the paper in strict accordance with the reviewers' suggestions. Editorial comments are listed below in red font. Reviewer comments are in blue font, such as below, where specific questions are numbered. Our responses are given in black font. The following are our responses and explanations to the reviewers' comments:

1.Please ensure that your manuscript meets PLOS ONE's style requirements, including those for ﬁle naming:

In order to comply with the strict formatting requirements of PLOS ONE, we have revised the formatting of the previous manuscript, which now conforms to PLOS ONE style, as described under "Revised Manuscript with Track Changes" and "Manuscript."

2.Please include captions for your Supporting Information files at the end of your manuscript, and update any in-text citations to match accordingly:

We have included the title of this research supporting information file at the end of the manuscript in strict accordance with the editor's request, and updated any textual citations accordingly,as described under "Revised Manuscript with Track Changes" and " Manuscript."

3.We note that your Data Availability Statement is currently as follows:”All relevant data are within the manuscript and its Supporting Information files.”Please confirm at this time whether or not your submission contains all raw data required to replicate the results of your study. Authors must share the “minimal data set” for their submission. PLOS ONE defines the minimal data set to consist of the data required to replicate all study findings reported in the article, as well as related metadata and methods:

In accordance with the requirements of PLOS ONE's Data Availability Statement and the recommendations of the editors, we have examined all of the raw data used in this study and assure you that our paper contains all of the raw data necessary to replicate the results of the study, as described in Table 1、Table 2 and Figure 1 of the manuscript and in the Supporting Information (S1 File、S2 File、S3 Fig、S4 Table.).

4.INTRODUCTION：There are some reviews about medical clowning for sick children, how does this article differ from them：

Thank you very much for your valuable comment. Based on your advice, we searched two other similar articles on clown care (https://doi.org/10.1007%2Fs12519-023-00720-y and https://doi.org/10.3389%2Ffped.2024.1324283). By carefully reading the full text, we have detailed the differences between the two articles and this study, as follows.

The purpose of article 1 (https://doi.org/10.1007%2Fs12519-023-00720-y) is to analyze the effects of clown care on hospitalized children. The findings suggest that clown care can reduce children's distress, relieve their anxiety, shorten their crying time, and raise caregivers' anxiety levels. However, some young children may be frightened by clowns and become tearful or anxious. The clown should be trained to notice and understand these children, and staff and parents should be consulted to help reduce anxiety until the particular child is satisfied with the clown. This is consistent with the results of this study. The purpose of article 2 (https://doi.org/10.3389%2Ffped.2024.1324283) was to evaluate the effectiveness of medical clowns in reducing pain and anxiety in hospitalized pediatric patients and their parents in different areas of medicine. The findings suggest that medical clowns have substantial positive and beneficial effects on reducing stress and anxiety in pediatric children and their families in a variety of settings, consistent with the findings of this study.

However, different from these two articles, the purpose of this study is to summarize the timing, form, and content of intervention, outcome indicators, and effects of clown care in hospitalized children. The first two articles only analyzed the application effect of clown care on hospitalized children but did not analyze the intervention timing, intervention form and content, and outcome indicators of clown care in different diseases. At the same time, the pain assessment in the above two articles may be limited to the pain assessment scale based on facial expression and behavior due to the incomplete literature included, while this study noted through literature analysis that the pain assessment tools based on facial expression and behavior may be limited to the measurement of subcortical body and autonomic motor pathways. This does not objectively diagnose the pain experience of the child. Cortisol is the main hormone in the body under stress, which can effectively detect pain in children , the research objects included in the first two articles are mostly children receiving invasive medical operations and surgeries, and this study also includes children with autism and children with cancer, so the research objects are more abundant, and the scope of clown care is wider.

In short, other articles on clown care mostly focused on analyzing the clinical application effect of clown care in hospitalized children. However, compared with other similar articles on clown care, this study not only analyzed the application effect of clown care but also fully analyzed and summarized its intervention timing, intervention form and content, and outcome indicators. In order to provide a reference for medical personnel to carry out relevant intervention.See "Manuscript with Track Changes" and "Manuscript".

5.INTRODUCTION：Why conduct a scoping review rather than other types of reviews：

We appreciate the editor's advice, which has been very helpful to us, and the following is what we added:

Many reviews have clarified the effectiveness of the application of clown care in pediatric patients, but there is a lack of comprehensive descriptions of the intervention timing, form and content of intervention, and outcome indicators of existing clown care. A scope review can summarize a range of documentary evidence to show the breadth and depth of a knowledge domain. Therefore, guided by the scope review report framework, this study summarized and analyzed the relevant literature in this field at home and abroad, and summarized the intervention timing, form and content of intervention, outcome indicators, and effects of clown care in hospitalized children, with a view to providing references for medical staff to carry out relevant research.

6.METHODS：For Eligibility criteria: Please state in PICOS：

Thank you very much for your valuable comment. Based on your advice, we changed the Eligibility criteria to PICOS format. Population (P): Hospitalized children < 18 years of age, along with any accompanying family members; Intervention (I): Receiving hospital clowns intervention; Comparison (C): Compared receipt of hospital clowns to standard of care; Outcome (O): The application and effect evaluation of clown care in hospitalized children; Study design (S): Original research literature, encompassing qualitative studies, case-control studies, cohort studies, case studies, etc. See the manuscript for details.

7.A similar meta-analysis review was conducted and completed in November 2019 and they yielded 14 studies. However, your review is a scoping review and should include more papers. Please revisit if you missed any potential research:

We strongly agree with the editor's view.In order to guarantee the scientific soundness and comprehensiveness of the scope review outcomes,we have strictly followed the editor's comments,and researched relevant studies on clown care and screened and read articles according to the inclusion and exclusion criteria in this paper. We have meticulously reviewed and incorporated several articles that are closely aligned with our research objectives and methodologies. We believe this will enhance the discourse and contextual framework of our study. Interestingly, we found two articles from other types of reviews, the last two papers in Table 2 ("Bertini et al[25], 2011"; "Shauna et al[26], 2011"). The research objects of these two articles have not been included in this study (the research objects are children hospitalized for respiratory pathologies and disabled children, respectively), and new information has been extracted from these two articles. For specifics, refer to the manuscript.We trust that the updated manuscript will fulfill your expectations.

8.METHODS：The discussion addresses the key points and limitations of the current study. However, it could be strengthened by 1) a more explicit comparison of the findings with the existing literature. 2) elaborate the implications for practice and policy; 3) provide more specific recommendations for future research:

We have meticulously updated the discussion section in strict alignment with the editor's recommendations and provided additional clarifications corresponding to each suggestion made by the editor. as described under "Revised Manuscript with Track Changes" and " Manuscript."

Reviewer 1:

Regarding Reviewer 1's comments, we are very grateful to the reviewer for taking the time to read our paper. We would like to thank you for your professional review work, constructive comments, and valuable suggestions on our manuscript. Your time and efforts are greatly appreciated. We have revised the manuscript according to your comments as listed in detail below.

1.Clearly state your research difference from other similar studies in 2023 and 2024(https://doi.org/10.1007%2Fs12519-023-00720-y, and https://doi.org/10.3389%2Ffped.2024.1324283):

Thank you very much for your valuable comment. Based on your advice,we have read both articles on clown care (https://doi.org/10.1007%2Fs12519-023-00720-y and https://doi.org/10.3389%2Ffped.2024.1324283). By carefully reading the full text, we have detailed the differences between the two articles and this study, as follows.

The purpose of article 1 (https://doi.org/10.1007%2Fs12519-023-00720-y) is to analyze the effects of clown care on hospitalized children. The findings suggest that clown care can reduce children's distress, relieve their anxiety, shorten their crying time, and raise caregivers' anxiety levels. However, some young children may be frightened by clowns and become tearful or anxious. The clown should be trained to notice and understand these children, and staff and parents should be consulted to help reduce anxiety until the particular child is satisfied with the clown. This is consistent with the results of this study. The purpose of article 2 (https://doi.org/10.3389%2Ffped.2024.1324283) was to evaluate the effectiveness of medical clowns in reducing pain and anxiety in hospitalized pediatric patients and their parents in different areas of medicine. The findings suggest that medical clowns have substantial positive and beneficial effects on reducing stress and anxiety in pediatric children and their families in a variety of settings, consistent with the findings of this study.

However, different from these two articles, the purpose of this study is to summarize the timing, form, and content of intervention, outcome indicators, and effects of clown care in hospitalized children. The first two articles only analyzed the application effect of clown care on hospitalized children but did not analyze the intervention timing, intervention form and content, and outcome indicators of clown care in different diseases. At the same time, the pain assessment in the above two articles may be limited to the pain assessment scale based on facial expression and behavior due to the incomplete literature included, while this study noted through literature analysis that the pain assessment tools based on facial expression and behavior may be limited to the measurement of subcortical body and autonomic motor pathways. This does not objectively diagnose the pain experience of the child. Cortisol is the main hormone in the body under stress, which can effectively detect pain in children , the research objects included in the first two articles are mostly children receiving invasive medical operations and surgeries, and this study also includes children with autism and children with cancer, so the research objects are more abundant, and the scope of clown care is wider.

In short, other articles on clown care mostly focused on analyzing the clinical application effect of clown care in hospitalized children. However, compared with other similar articles on clown care, this study not only analyzed the application effect of clown care but also fully analyzed and summarized its intervention timing, intervention form and content, and outcome indicators. In order to provide a reference for medical personnel to carry out relevant intervention.We also present the results of the analysis in the manuscript. See the manuscript for details.

Reviewer 2:

Regarding Reviewer 2's comments, we would like to thank the reviewer for taking the time to read our paper and for giving us the opportunity to revise it with many meaningful comments. We have carefully read Reviewer 2's comments and revised each of them in order to take this opportunity to improve the rigor and scientific quality of our paper, as shown in the following responses:

1.Please rephrase this as PLOS Guidelines using the format "Department, Institution, City, State, Country". Adjust font, line spacing and other formatting:

We've rephrased it as the PLOS Guidelines as "Department, Agency, City, State, country,"As follows: School of Nursing, Yunnan University of Chinese Medicine, Kunming, Yunnan, China; Pediatric, Yan'an Hospital Affiliated To Kunming Medical University, Kunming, Yunnan, China.

At the same time, we also reset the font, line spacing, and other formats in strict accordance with the reviewer's annotations and PLOS ONE requirements.

2.ABSTRACT：Please attempt to shorten abstract by following the PLOS guidelines:

 Thanks to the reviewer's reminder, We have shortened the abstracts in strict accordance with the requirements of the PLOS guidelines.Please refer to the "Manuscript with Track Changes" and "Manuscript." 

3.ABSTRACT:（“Research on the efficacy of clown care for hospitalized children has produced varying results, with differences in treatment duration and methods.” ）This is an important point, but it lacks specificity. What kind of variability in the results has been observed (e.g., effectiveness across age groups or types of conditions)? Clarifying this would help the reader understand the motivation behind conducting a scoping review.

We sincerely appreciate the suggestions provided by the reviewer and have implemented changes based on their feedback. However, due to the constraints of the abstract, we will relocate the enhanced specific content to the introduction section, where it will be more detailed. Please refer to the "Manuscript with Track Changes" and "Manuscript." 

4.ABSTRACT：(“A total of 16 articles were included”)The number of articles included in the review is mentioned, but there is no indication of the total number of articles screened or excluded. Including these numbers would provide a better sense of the comprehensiveness of the review process. For example: "Out of X articles screened, 16 were included.":

Thanks to the reviewers' comments, We've updated it to reflect the reviewers' feedback and now read, "Out of 1084 articles screened, 18were included."

5.ABSTRACT：(“Clown care can be used with children who have burns, perioperative care, fracture rehabilitation, autism, puncture therapy, and chemotherapy, according to a review of the literature.”) This sentence effectively summarizes the range of conditions where clown care is applied. However, the results could be enhanced by specifying which conditions showed the most positive outcomes or where clown care was less 

---

## [Editor Report · Decision Letter 1]

29 Oct 2024

PONE-D-24-35005R1Application of clown care in hospitalized children: A scoping reviewPLOS ONE

Dear Dr. Hu,

Thank you for submitting your manuscript to PLOS ONE. After careful consideration, we feel that it has merit but does not fully meet PLOS ONE’s publication criteria as it currently stands. Therefore, we invite you to submit a revised version of the manuscript that addresses the points raised during the review process.

We look forward to receiving your revised manuscript.

Kind regards,

Cho Lee Wong, PhD

Academic Editor

PLOS ONE

Journal Requirements:

Additional Editor Comments :

Dear Authors

Thank you for your revision. Please shorten your conclusion and place it under the heading "Implications."

---

## [Author Response · Author response to Decision Letter 1]

30 Oct 2024

Dear Editors:

Greetings!

First of all, on behalf of all the authors, please allow me to extend our sincerest thanks to you and the reviewers. We would like to thank you for taking the time out of your busy schedules to review our paper in detail and provide valuable comments and suggestions. We are fully aware of the importance of these comments in improving the quality of the paper and ensuring academic rigor.

After receiving the review comments, we immediately organized our team to conduct in-depth discussion and analysis, and revised and improved the paper in strict accordance with the editors' suggestions. Editorial comments are listed below in red font. such as below, where specific questions are numbered. Our responses are given in black font. The following are our responses and explanations to the editorss' comments:

1.Please review your reference list to ensure that it is complete and correct. If you have cited papers that have been retracted, please include the rationale for doing so in the manuscript text, or remove these references and replace them with relevant current references. Any changes to the reference list should be mentioned in the rebuttal letter that accompanies your revised manuscript. If you need to cite a retracted article, indicate the article’s retracted status in the References list and also include a citation and full reference for the retraction notice:

I sincerely appreciate your guidance. At the same time, we sincerely apologize for any inconvenience our errors may have caused you. After receiving your recommendation, we looked over the references right away and, per your request, deleted the ones that had been withdrawn and substituted the most recent ones. Again, I apologize. Please refer to the "Manuscript with Track Changes" and "Manuscript." 

2.Thank you for your revision. Please shorten your conclusion and place it under the heading "Implications.":

I appreciate your suggestion very lot. In response to your suggestion, we have condensed the conclusion and placed it under "Implication".Please refer to the "Manuscript with Track Changes" and "Manuscript." 

Once again, I would like to thank the editor and reviewers for their comments and giving me the opportunity to revise my paper. Through this thesis revision process, I deeply realized the shortcomings in our research and made comprehensive and in-depth improvements under your guidance. I believe that after these revisions, the quality of the thesis has been significantly improved and is more in line with the publication standards of PLOS ONE journals. Once again, I would like to thank you for your hard work and valuable comments and look forward to your further review and affirmation.

---

## [Editor Report · Decision Letter 2]

1 Nov 2024

Application of clown care in hospitalized children: A scoping review

PONE-D-24-35005R2

Dear Dr. Hu,

We’re pleased to inform you that your manuscript has been judged scientifically suitable for publication and will be formally accepted for publication once it meets all outstanding technical requirements.

Kind regards,

Cho Lee Wong, PhD

Academic Editor

PLOS ONE
---

## [Editor Report · Acceptance letter]

26 Nov 2024

PONE-D-24-35005R2 

PLOS ONE

Dear Dr. Xue, 

I'm pleased to inform you that your manuscript has been deemed suitable for publication in PLOS ONE. Congratulations! Your manuscript is now being handed over to our production team.

Kind regards, 

on behalf of

Dr. Cho Lee Wong 

Academic Editor

PLOS ONE